# Distribution of nutrients and dissolved organic matter in a eutrophic equatorial estuary: the Johor River and the East Johor Strait

**Amanda Y. L. Cheong**[1,a], **Kogila Vani Annammala**[2,3], **Ee Ling Yong**[2,4], **Yongli Zhou**[1,b], **Robert S. Nichols**[1,c], and **Patrick Martin**[1]

[1]Asian School of the Environment, Nanyang Technological University, 639798, Singapore

[2]Department of Water and Environmental Engineering, Faculty of Civil Engineering, Universiti Teknologi Malaysia, 81310 Johor Bahru, Johor, Malaysia

[3]Disaster Preparedness and Prevention Centre (DPPC), Malaysia–Japan International Institute of Technology (MJIIT), Universiti Teknologi Malaysia, 54100 Kuala Lumpur, Malaysia CE1

[4]Centre for Environmental Sustainability and Water Security, Universiti Teknologi Malaysia, 81310 Johor Bahru, Johor, Malaysia

[a]present address: Aon Singapore Pte Ltd., 068804, Singapore

[b]present address: Department of Geography, University of Hong Kong, Hong Kong SAR, China

[c]present address: DHI Water & Environment (S) Pte Ltd., 608526, Singapore

**Correspondence:** Patrick Martin (pmartin@ntu.edu.sg)

**Abstract.** Estuaries have strong physicochemical gradients that lead to complex variability and often high rates of biogeochemical processes, and they are also often impacted by humans. Yet, our understanding of estuarine biogeochemistry remains skewed towards temperate latitudes. We examined seasonal and spatial variability in dissolved organic matter (DOM) and nutrients along a partly eutrophic, agricultural–urban estuary system in Southeast Asia: the Johor River and the East Johor Strait CE2. Dissolved organic carbon (DOC) and coloured DOM (CDOM) showed non-conservative mixing, indicating significant DOM inputs along the estuary. The CDOM spectral slopes and CDOM : DOC ratios suggest that terrigenous, soil-derived DOM dominates along the Johor River, while phytoplankton production and microbial recycling are important DOM sources in the Johor Strait. CDOM properties were not unambiguous source indicators in the eutrophic Johor Strait, which is likely due to heterotrophic CDOM production. Nitrate concentrations showed conservative mixing, while nitrite concentrations peaked at intermediate salinities of 10–25. Ammonium concentrations decreased with salinity in the Johor River but increased up to $50\,\mu mol\,L^{-1}$ in the Johor Strait, often dominating the dissolved inorganic nitrogen (DIN) pool. Phosphate concentrations were low ($< 0.5\,\mu mol\,L^{-1}$) throughout the Johor River but increased in the Johor Strait, where DIN : phosphate ratios were typically $\geq 16:1$. This suggests that the Johor Strait may experience phosphorus limitation and that internal recycling is likely important for maintaining high nutrient concentrations in the Johor Strait. Overall, our results indicate that the Johor River and Johor Strait are clearly not part of the same estuarine mixing continuum and that nutrient recycling processes must be quantified to understand nutrient dynamics in the Johor Strait. Moreover, our results highlight the need for better techniques for DOM source tracing in eutrophic estuaries.

## 1 Introduction

The biogeochemical functioning of estuaries and coastal waters is greatly influenced by terrestrial inputs and by biogeochemical transformations taking place along the land–ocean aquatic continuum (Bianchi and Morrison, 2023; Martin and Bianchi, 2024; Voss et al., 2011). These fluxes and biogeochemical processes can be greatly affected by increasing coastal development, changing land-use practices, and

climatic changes. Because of the ecological and economic value of estuaries and coastal waters, it is important to better understand these fluxes and processes and how they are changing. Southeast Asia is experiencing some of the highest rates of coastal urbanisation (Neumann et al., 2015) and land-use change (Hansen et al., 2013; Stibig et al., 2014) as well as increased nutrient pollution (Sinha et al., 2019). However, our understanding of estuarine and coastal biogeochemistry remains skewed towards temperate latitudes (Lønborg et al., 2021b; Vieillard et al., 2020). Thus, although tropical estuaries receive a large fraction of the global land–ocean fluxes of carbon, nutrients, and sediments (Jennerjahn, 2012), many tropical estuaries are comparatively poorly studied.

Eutrophication of estuaries and coastal waters due to anthropogenic nutrient input is a worldwide problem, resulting in phytoplankton blooms, oxygen depletion, and dead zones (Altieri et al., 2017; Le Moal et al., 2019). Globally, agriculture is the main source of anthropogenic N and P input, but many other point and non-point sources may be important in a given location (Beusen et al., 2016; Le Moal et al., 2019). Anoxic conditions in eutrophic systems can promote nutrient recycling, especially the release of phosphorus from sediments (Sulu-Gambari et al., 2018; Ballagh et al., 2020). Although nitrogen is usually lost as a result of denitrification and anaerobic ammonia oxidation (anammox) under anoxic conditions (Voss et al., 2011; Zhu et al., 2013; Teixeira et al., 2016), dissimilatory nitrate reduction to ammonia (DNRA) can also take place and recycle nitrogen (Dong et al., 2011; Bernard et al., 2015; Chai et al., 2021). Importantly, the relative rates of these biogeochemical processes may differ between tropical and temperate systems (Dong et al., 2011; Li et al., 2019).

Estuaries also receive large fluxes of terrestrial organic carbon, partly as a result of human activities (Regnier et al., 2022; Martin and Bianchi, 2024). Tropical rivers are particularly significant sources of dissolved organic carbon (DOC) to the ocean (Dai et al., 2012), with mangroves thought to be a disproportionally large source of terrigenous DOC (Dittmar et al., 2006). Terrigenous DOC is typically rich in coloured dissolved organic matter (CDOM) (Coble, 2007; Massicotte et al., 2017), and terrigenous CDOM typically has distinct optical properties compared to CDOM in the open ocean and coastal seas (Stedmon and Nelson, 2015). Specifically, the CDOM spectral slopes at ultraviolet wavelengths (Helms et al., 2008) and specific UV absorbance at 254 nm, $SUVA_{254}$ (Traina et al., 1990; Weishaar et al., 2003), have become widely used metrics for distinguishing terrigenous dissolved organic matter (DOM) from autochthonous DOM produced in aquatic environments (Asmala et al., 2016; Fichot and Benner, 2011; Lønborg et al., 2021a; Zhou et al., 2021).

Tropical peatlands are the largest source of terrigenous DOC to the coastal ocean in Southeast Asia (Alkhatib et al., 2007; Baum et al., 2007; Moore et al., 2011). Most research on land–ocean DOC fluxes in this region has therefore focused on peatland-draining rivers (Baum et al., 2007; Wit et

al., 2015; Martin et al., 2018; Rixen et al., 2022; Sanwlani et al., 2022), leaving us with a more limited understanding of the concentrations and optical properties of DOM in non-peat-draining estuaries. Moreover, there have been few studies of the distributions of DOM and nutrients across more urbanised and eutrophic estuaries in Southeast Asia.

Tanaka et al. (2021) recently reviewed anthropogenic impacts on tropical river and estuary biogeochemistry. In general, inorganic nutrient inputs are increased, while nitrogen-to-phosphorus (N : P) ratios and organic carbon fluxes can increase or decrease depending on the dominant anthropogenic factors (Tanaka et al., 2021). However, many site-specific factors, such as water residence times and specific land management practices, greatly control the biogeochemistry of any specific location (Tanaka et al., 2021). For example, the Can Gio estuary downstream of Ho Chi Minh City (Vietnam) receives high inputs of wastewater nutrients, but the inorganic N pool is dominated by nitrate (because wastewater ammonia is nitrified within the estuary), and dissolved inorganic N : P ratios are generally low (Taillardat et al., 2020). In contrast, the Klang River estuary downstream of Kuala Lumpur (Malaysia) has an inorganic N pool dominated by ammonia from riverine inputs, but peaks in ammonia concentration during periods of high river flow are not always accompanied by peaks in phosphate concentration, leading to very variable N : P ratios (Lim et al., 2019; Lee et al., 2020). Jakarta Bay (Indonesia) also receives high nutrient inputs from urban wastewater via rivers, with low N : P ratios in river water and hypereutrophic conditions near to the shore. However, here, the physical ocean circulation disperses the nutrient input and results in very strong horizontal gradients across the bay (van der Wulp et al., 2016; Damar et al., 2020).

Here, we examined the seasonal dynamics and mixing behaviours of DOC, CDOM, and dissolved inorganic nutrients across salinity gradients in the Johor River estuary and the eutrophic East Johor Strait, located at the southern tip of the Malay Peninsula. The biogeochemistry of this system has so far received only very limited attention and has not been the focus of dedicated studies. The objectives of our research were to identify the sources (terrigenous or marine) and cycling behaviour of these substances, determine whether there is seasonal variation, and examine how the optical properties of CDOM delivered by the Johor River compare to those of the eutrophic Johor Strait.

## 2 Methods

### 2.1 Study area

The Johor River (Fig. 1) is around 123 km long and provides a crucial freshwater source for the state of Johor and for Singapore (Kang and Kanniah, 2022). The Johor River drains into a large estuary (with a total drainage basin area of around 2640 km$^2$) to which several other rivers also con-

tribute, before flowing southwards past the East Johor Strait into the Singapore Strait (Fig. 1). The land cover within the catchment is predominantly agricultural, primarily comprising rubber and oil palm plantations, with some urban and industrial land use (Kang and Kanniah, 2022; Fig. 1a). The Johor River estuary has fringing mangroves along most of its length, providing a narrow buffer zone for the mostly agricultural areas. Sand extraction takes place in the river, and in the lower parts of the estuary, small-scale aquaculture is practised. Along the Johor River and its estuary, wastewater treatment plants and fertiliser run-off are sources of ammonia, with point-source inputs via wastewater being more important during drier periods and non-point-source inputs from fertiliser use being more important during wetter periods (Samsudin et al., 2017; Pak et al., 2021).

The Johor Strait is a narrow (1–2 km wide) channel that separates Singapore from Malaysia (Fig. 1a and b). It is divided by a causeway into the East and West Johor straits, with limited water exchange between the two. The East Johor Strait receives run-off from both Malaysia and Singapore, with both sides of the strait dominated by urban and industrial land use. Fringing mangroves are also found, especially in the eastern part of the strait (Fig. 1a). Aquaculture is practised, especially around the island of Pulau Ubin, which is mostly a forested nature reserve. Both the East and West Johor straits are eutrophic water bodies, and occasional harmful algal blooms caused by diatoms and dinoflagellates have been reported (Gin et al., 2000; Chénard et al., 2019; Kok and Leong, 2019; Wijaya et al., 2023).

Singapore and Malaysia experience two monsoon seasons: the northeast monsoon from November to March and the southwest monsoon from mid-May to mid-September. Rain falls all throughout the year, but there is a distinct increase in precipitation during the early northeast monsoon (mid-November to early January), followed immediately by the driest time of year in the late northeast-monsoon period (February to March). Average monthly rainfall for 2000–2020 and monthly rainfall during our study period were calculated over a defined area (1.2° N, 103.6° E–1.8° N, 104.4° E) using data from the IMERG (Integrated Multi-Satellite Retrievals for Global Precipitation Measurement) Final Precipitation L3 product (0.1° horizontal resolution) processed by the Royal Netherlands Meteorological Institute (Huffman et al., 2014). Discharge from the Johor River at the gauging station in Rantau Panjang (1.781° N 103.746° E) was obtained from the Department of Irrigation and Drainage, Malaysia. We designated data collected prior to December 2017 as "pre-monsoon", data collected during December and January as "monsoon", and data collected after January 2018 as "post-monsoon" and distinguished between these three periods in our plotting and analysis.

## 2.2 Sampling

We collected surface water samples at eight stations in the East Johor Strait, approximately monthly, from August 2017 to June 2018 (Fig. 1c). Samples were collected from the Johor River on 3–4 November 2017 (pre-monsoon, i.e. sampled before the monsoon peak in river discharge; see Fig. 2), 10 January 2018 (monsoon), and 6 March 2018 (post-monsoon) at 8–10 stations (Fig. 1b). Water samples were collected either using a handheld polyethylene jug (Johor River) or a Niskin bottle (Johor Strait) at 1 m depth. At each station, conductivity–temperature–depth (CTD) and chlorophyll-*a* (chl-*a*) fluorescence profiles were taken using a FastCTD, and we used the average values over the upper 1 m in our analysis. Salinity is reported using the Practical Salinity Scale and is therefore unitless. Due to a CTD malfunction on 9 March 2018, CTD profiles could not be taken in the Johor Strait on this date, and salinity was instead measured using an optical refractometer.

Samples for dissolved inorganic nutrients were syringe-filtered (0.22 μm, Pall Acrodisc) in the field into acid-rinsed 15 mL polypropylene centrifuge tubes, immediately frozen in a dry shipper, and then stored at $-20\,^{\circ}$C until analysis. For DOC and CDOM, unfiltered water was stored in pre-combusted (450 $^{\circ}$C, 4 h) amber borosilicate bottles at ambient temperature in the dark and then filtered back on land on the same day through Whatman Anodisc filters (0.2 μm, 45 mm diameter) in an all-glass filtration system. The filtration system was rinsed with 1 M HCl and ultrapure deionised water (ELGA LabWater, 18.2 M$\Omega$ cm$^{-1}$), hereafter referred to as DI water, prior to every sample. DOC and CDOM samples were then stored in amber borosilicate vials at $+4\,^{\circ}$C until analysis; DOC samples were acidified with 50 % H$_2$SO$_4$ immediately after filtration.

## 2.3 Sample analyses

### 2.3.1 Dissolved organic carbon (DOC) analysis

Samples were analysed within 3 months of collection as non-purgeable organic carbon using a Shimadzu TOC-L system equipped with a Shimadzu high-salt kit, with potassium hydrogen phthalate used for calibration. Instrumental detection limits were below the lowest standard concentration of $\sim 20\,\mu$mol L$^{-1}$ and thus much lower than the lowest measured sample of 71 μmol L$^{-1}$. Deep-seawater certified reference material from the University of Miami (42–45 μmol L$^{-1}$) was analysed with every run and returned a long-term mean and standard deviation of $48 \pm 3.5\,\mu$mol L$^{-1}$ during the period 2017–2018.

### 2.3.2 Coloured dissolved organic matter (CDOM) analysis

CDOM absorption spectra (230–900 nm) were measured with a Thermo Scientific Evolution 300 dual-beam

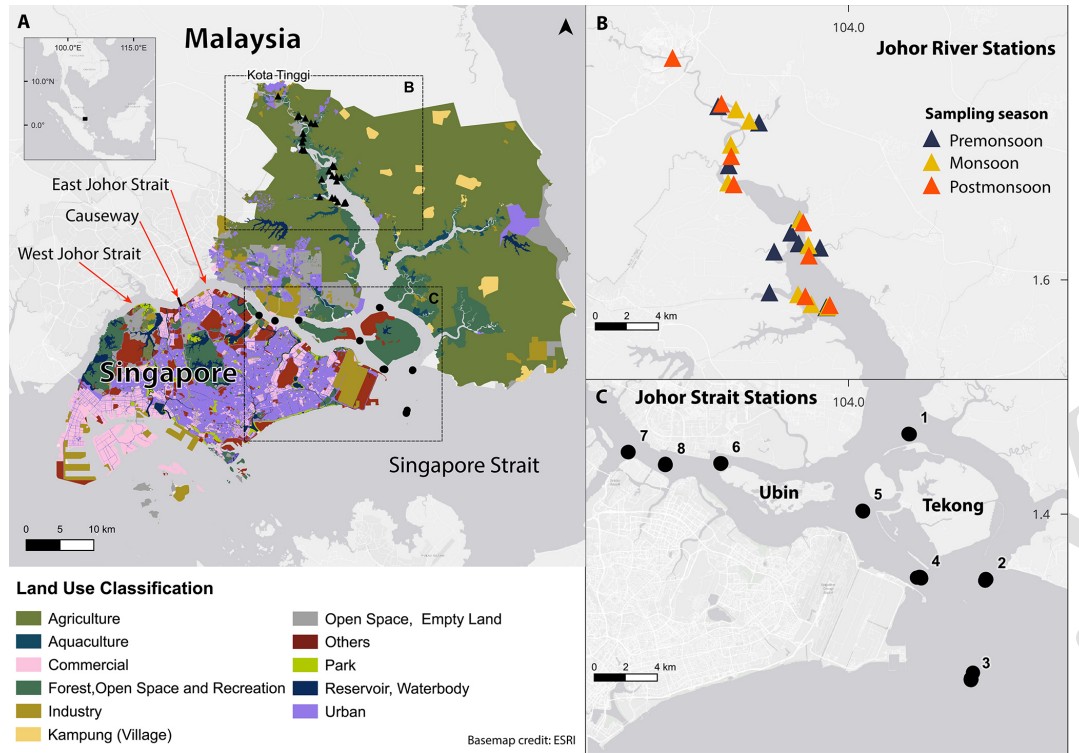

**Figure 1.** Map of the study region. Triangles show stations in the Johor River estuary; points show stations in the East Johor Strait. **(a)** Land use in the area around the Johor River estuary and East Johor Strait. Land-use data for Malaysia were taken from the GeoJohor portal (which focuses on Johor land use) (http://geoportal.johor.gov.my/en/, last access: 11 December 2022); land-use data for Singapore were taken from the Singapore Urban Redevelopment Authority Master Plan (https://www.ura.gov.sg/maps/?service=mp, last access: 10 January 2024). Dashed boxes show the locations for panels **(b)** and **(c)**. **(b)** Station locations in the Johor River estuary at which sampling took place. **(c)** Station locations in the East Johor Strait at which sampling took place.

spectrophotometer against a DI-water reference using 10 cm quartz cuvettes. The absorbance spectra were corrected for instrument baseline drift following Green and Blough (1994), smoothed using a LOESS function and converted to Napierian absorption coefficients using Eq. (1),

$$a_\lambda = 2.303 \times \frac{A_\lambda}{l}, \tag{1}$$

where $a_\lambda$ is the absorption coefficient ($m^{-1}$), $A_\lambda$ is the absorbance, $l$ is the cuvette path length (m), and the subscript $\lambda$ indicates wavelength. We then calculated the spectral slopes between 275–295 nm ($S_{275-295}$) and 350–400 nm ($S_{350-400}$), as well as the slope ratio $S_R$ (ratio of $S_{275-295}$ to $S_{350-400}$) using linear regressions of the natural log-transformed absorption against wavelength, following Helms et al. (2008). The specific UV absorbance at 254 nm ($SUVA_{254}$) was determined by dividing the decadic absorption at 254 nm (i.e. the absorbance per metre) by the DOC concentration (in $mg\,L^{-1}$). CDOM data processing was carried out in MAT-LAB. We report CDOM absorption at 350 nm ($a_{350}$) as a measure of CDOM concentration.

### 2.3.3 Nutrient analysis

Samples for $NO_x$ (i.e. $NO_3^- + NO_2^-$), $NO_2^-$, $PO_4^{3-}$, $Si(OH)_4$, and $NH_4^+$ were thawed and analysed using a segmented-flow analyser (SEAL Analytical AutoAnalyzer 3 (AA3) HR) following SEAL methods G172, G173, G297, and G177. $NH_4^+$ was measured fluorometrically (Kérouel and Aminot, 1997). The detection limits were $0.05\,\mu mol\,L^{-1}$ ($NO_x$), $0.01\,\mu mol\,L^{-1}$ ($NO_2^-$), $0.03\,\mu mol\,L^{-1}$ ($PO_4^{3-}$), $0.10\,\mu mol\,L^{-1}$ ($Si(OH)_4$), and $0.25\,\mu mol\,L^{-1}$ ($NH_4^+$). Dissolved inorganic nitrogen (DIN) was calculated as $NO_x + NH_4^+$. Where measurements were below the detection limit (1 sample for $PO_4^{3-}$ and 25 samples for $NH_4^+$), the concentration was assumed to be 0.5 times the detection limit. The nutrient data for the Johor River samples were previously published by Liang et al. (2020).

### 2.4 Conservative mixing models and statistical analysis

For DOC and CDOM, we used a two-endmember mixing model to calculate the concentrations expected under conservative mixing across the salinity gradient. The fresh-

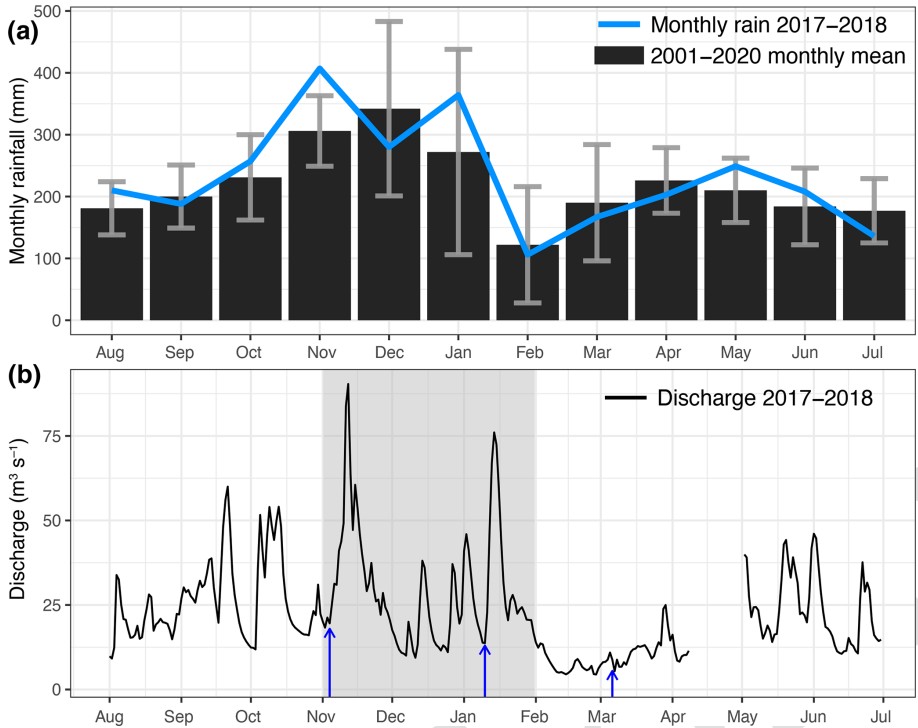

**Figure 2. (a)** Monthly rainfall across the study region during the period August 2017 to July 2018 (blue line) and long-term monthly-mean rainfall from 2001–2020 (black bars). Grey error bars each indicate 1 standard deviation of the long-term mean. **(b)** Daily river discharge for the Johor River measured at the Rantau Panjang hydrological station (1.781° N, 103.746° E). Blue arrows indicate the Johor River sampling dates – for pre-monsoon sampling, the arrow denotes 4 November.

water endmember values were taken from the station furthest upstream in the Johor River (Kota Tinggi: 1.6973° N, 103.9358° E), while the marine endmember values were taken from the southernmost station (station 3 in Fig. 1c – where the Johor Strait opens into the Singapore Strait). The mixing models were calculated for DOC and for CDOM absorption spectra using Eq. (2),

$$X_{\mathrm{mix}} = f_{\mathrm{riv}} X_{\mathrm{riv}} + (1 - f_{\mathrm{riv}}) X_{\mathrm{mar}}, \tag{2}$$

where $X_{\mathrm{mix}}$ is the predicted DOC concentration or CDOM absorption at each intermediate salinity, $f_{\mathrm{riv}}$ is the fraction of freshwater at that salinity, and $X_{\mathrm{riv}}$ and $X_{\mathrm{mar}}$ are the riverine- and marine endmember values for DOC or CDOM absorption. The mixing models were calculated using salinity increments of 1. Because the CDOM spectral slopes change nonlinearly during conservative mixing, we calculated the predicted absorption spectrum at every wavelength using Eq. (2) and then calculated the predicted spectral-slope parameters from the predicted spectra, following Stedmon and Markager (2003).

Where two variables clearly showed a coherent and linear relationship to each other, we used linear regression analysis to test for a significant relationship and reported the regression equation together with $r^2$ and $p$ values. In cases where there was greater scatter or visible non-linearity, we used Spearman's rank correlation analysis instead and reported the $\rho$ values and $p$ values.

## 3 Results

### 3.1 Rainfall pattern and river discharge in Kota Tinggi, Johor

Rainfall across the study region (Fig. 2a) is relatively high all throughout the year but exhibits a distinctly wetter period during the early part of the northeast monsoon (November–January) and a drier period during the late northeast monsoon (February–March). During our study period, November 2017 was the wettest month with 407 mm of rainfall, while February 2018 was the driest month with 106 mm of rainfall. The overall monthly rainfall distribution during our study period was similar to the long-term average from 2001 to 2020.

Discharge of the Johor River measured at Rantau Panjang showed a distinct minimum during the dry period in the late northeast monsoon (February–March 2018; Fig. 2b). Discharge during the early northeast monsoon showed two distinct peaks, on 12 November 2017 (90.4 m³ s⁻¹) and 14 January 2018 (76.0 m³ s⁻¹), but otherwise, discharge was not notably elevated during November–January relative to other periods, except when compared to the late northeast

monsoon (February–March; Fig. 2b). Note that the pre-monsoon sampling from the Johor River was conducted on 3–4 November 2017, just before the river discharge began to increase (Fig. 2b).

## 3.2 Temperature and salinity

In the Johor River, surface salinity ranged from 0.5–26.8 (November, i.e. pre-monsoon), 0–16.1 (January, i.e. monsoon), and 4–28.3 (March, i.e. post-monsoon), with the lowest salinity always observed furthest upstream in the town of Kota Tinggi (Figs. S1–S3 in the Supplement). Salinity ranged from 21.6–32.4 for all stations in the Johor Strait, with the lowest salinity (mean: 25.9; range: 21.6–29.0) typically found at station 1, where the Johor River estuary meets the Johor Strait, while station 3, located at the entrance to the Singapore Strait, had the highest salinity (mean: 30.9; range: 30.0–32.4).

Surface temperature varied little and spanned nearly identical ranges in the Johor River (26.7–30.2 °C) and the Johor Strait (26.5–31.6 °C), with lower values observed during the northeast monsoon but otherwise no distinct seasonal variation noted.

## 3.3 Dissolved organic matter distribution

DOC concentrations ranged from 101 to 186 $\mu$mol L$^{-1}$ in the Johor River (average: 136 $\mu$mol L$^{-1}$) and from 71 to 208 $\mu$mol L$^{-1}$ in the Johor Strait (average: 113 $\mu$mol L$^{-1}$; Fig. 3a). DOC concentrations in the Johor River estuary showed a clear non-conservative mixing pattern, indicative of additional DOC sources to the estuary, and only showed a decrease after salinity exceeded about 22. In the Johor Strait, DOC concentrations showed a relatively linear decrease with salinity, with the lowest values consistently found at Johor Strait station 3 (average of 79 $\mu$mol L$^{-1}$), closest to the open sea in the Singapore Strait (see Fig. 1). Seasonal variability was not pronounced, although at salinities between 5 and 25, DOC concentrations were lower during the northeast monsoon. Moreover, five stations in the Johor Strait showed elevated DOC concentrations relative to their salinity (four stations in February 2018 and one station in June 2018), which were associated with phytoplankton blooms (see Sect. 3.3). Overall, the distribution of DOC did not follow simple two-endmember conservative mixing between the Johor River and the Singapore Strait: essentially all stations along the salinity gradient had DOC concentrations that were higher than predicted (Fig. 3a).

Both CDOM $a_{350}$ and SUVA$_{254}$ showed a very similar distribution to DOC, with relatively stable values of up to a salinity of 25 in the Johor River, followed by a linear decrease with salinity above 25. CDOM $a_{350}$ ranged from 1.35–2.88 m$^{-1}$ in the Johor River and from 0.25–1.90 m$^{-1}$ in the Johor Strait (Fig. 3b), while SUVA$_{254}$ ranged from 2.1–3.2 L mg$^{-1}$ m$^{-1}$ throughout the Johor River

and from 0.8–3.0 L mg$^{-1}$ m$^{-1}$ in the Johor Strait (Fig. 3c). The CDOM spectral slope $S_{275-295}$ showed low values (0.0141–0.0176 nm$^{-1}$) at stations where salinity was < 30 but increased to 0.0196–0.0239 nm$^{-1}$ at the highest salinities (Fig. 3d). The spectral slope, $S_{350-400}$, showed a steadily decreasing trend over the salinity gradient – from $\geq 0.0194$ nm$^{-1}$ at salinities below 7 to values ranging between 0.0146–0.0188 nm$^{-1}$ in the Johor Strait (Fig. 3e). The slope ratio, $S_R$, consequently mirrored the pattern in $S_{275-295}$, with values consistently below 1.0 up to a salinity of 25 and then increasing to between 1.0–1.6 at the higher salinities (Fig. 3f). There was only limited seasonal variability in the CDOM parameters: values of $a_{350}$, $S_{275-295}$, and $S_{350-400}$ were slightly lower during the northeast monsoon at salinities between 5–25 than in the other seasons, but this was not seen in $S_R$ or in SUVA$_{254}$. $S_{275-295}$ clearly departed from the conservative mixing models, while $S_{350-400}$ and $S_R$ showed somewhat closer agreement (Fig. 3). The five stations with high DOC concentrations due to phytoplankton blooms did not stand out clearly in the CDOM parameters.

## 3.4 Chlorophyll-*a* concentration

The chl-*a* concentration, as measured with the CTD fluorometer, ranged mostly between 0–7 $\mu$g L$^{-1}$ TSI in both the Johor River and the Johor Strait, with the exception of five samples in the Johor Strait that had concentrations of 10.1–50.3 $\mu$g L$^{-1}$ (Fig. 4a). These were samples taken at stations 5, 6, 7, and 8 in February 2018 and at station 7 in June 2018, indicating that two phytoplankton blooms took place in the inner part of the Johor Strait. Excluding these five stations, the chl-*a* concentration showed no seasonal pattern in the Johor River or the Johor Strait.

The chl-*a* concentration was not significantly correlated with DOC concentration in the Johor River. In the Johor Strait, the concentration of DOC showed a clear linear relationship with chl-*a* concentration for the five bloom stations at which chl-*a* concentration exceeded 10 $\mu$g L$^{-1}$ ($y = 1.95x + 105.4$, $r^2 = 0.92$, $p = 0.01$). Excluding the five bloom stations, there was still a significant correlation between DOC and chl-*a* concentrations in the Johor Strait ($\rho = 0.41$, $p = 0.001$; Fig. 4b). This pattern was not seen with $a_{350}$ or with $S_{275-295}$ (no significant correlations), and the five bloom stations had $a_{350}$ values that were close to the overall mean $a_{350}$ for the Johor Strait (1.19 $\pm$ 0.39 m$^{-1}$) but had consistently low $S_{275-295}$ (Fig. 4c and d). However, the bloom stations showed a decreasing trend of SUVA$_{254}$ with chl-*a* concentration (although this was not statistically significant) and had relatively low SUVA$_{254}$ compared to the other Johor Strait data (Fig. 4e).

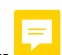

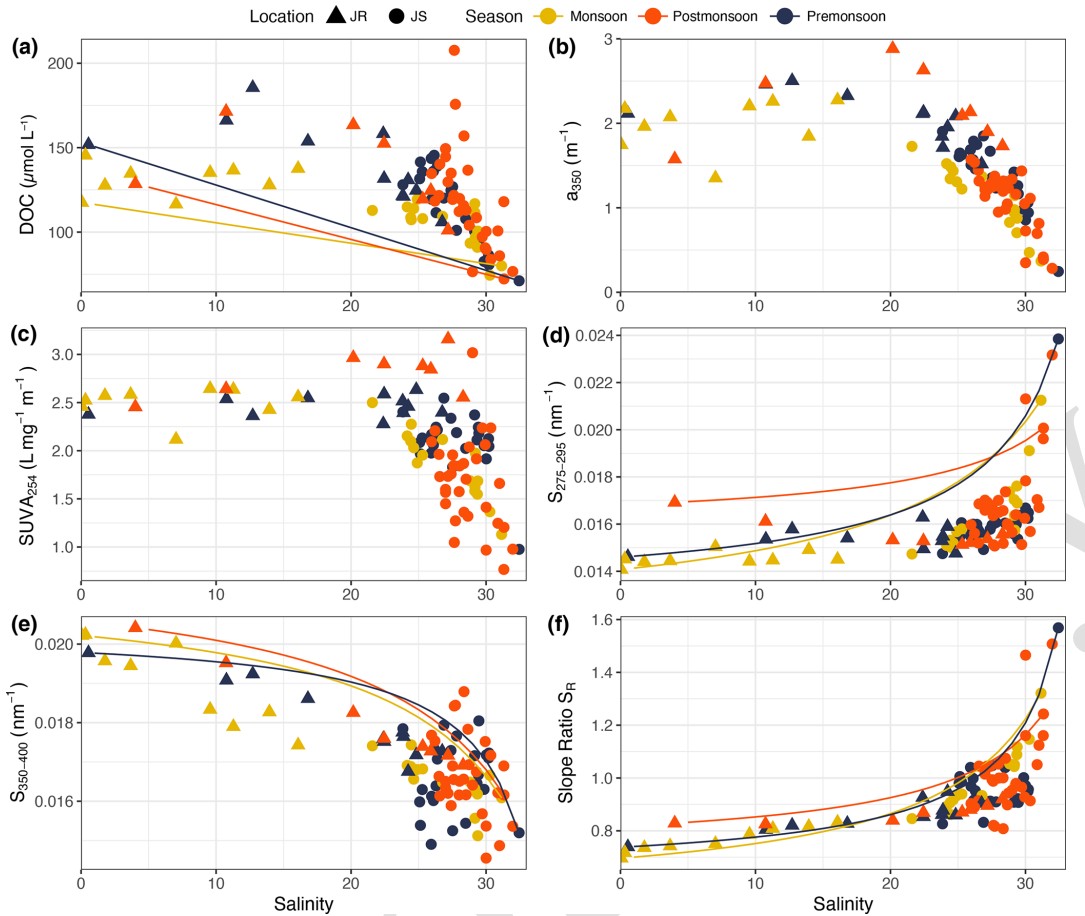

**Figure 3.** Distribution against salinity of **(a)** DOC concentration, **(b)** CDOM $a_{350}$, **(c)** SUVA$_{254}$, **(d)** the CDOM spectral slope $S_{275-295}$, **(e)** the CDOM spectral slope $S_{350-400}$, **(f)** and the CDOM spectral-slope ratio $S_R$. The triangles represent data from the Johor River (JR), circles represent data from the Johor Strait (JS), and symbol colours indicate the sampling season. Solid lines indicate the theoretical conservative mixing lines calculated for each sampling season.

## 3.5 Relationships between dissolved organic matter parameters

There was a significant correlation between $a_{350}$ and DOC concentration across all sampling seasons and stations (Spearman's rank correlation: $\rho = 0.69$, $p < 0.001$; Fig. 5a), although the Johor River stations typically had higher $a_{350}$ at a given DOC concentration than the Johor Strait stations. The CDOM spectral slope $S_{275-295}$ and the slope ratio $S_R$ both showed negative correlations with DOC concentration ($\rho = -0.40$ ($p < 0.001$) and $\rho = -0.49$ ($p < 0.001$), respectively; Fig. 5b and c). A strong inverse correlation was observed between $S_{275-295}$ and SUVA$_{254}$ (Spearman's rank correlation: $\rho = -0.71$, $p < 0.001$; Fig. 5d). The five Johor Strait bloom stations mostly had lower $a_{350}$ and SUVA$_{254}$ relative to the amount of DOC compared to the other data but did not have notably different $S_{275-295}$ or $S_R$ (compare Fig. 5a and d to Fig. 5b and c).

## 3.6 Dissolved inorganic nutrients

$NO_3^-$ concentration showed a strong and linear decrease with salinity ($y = -1.67x + 51.6$, $r^2 = 0.91$, $p < 0.001$), ranging from 3.1–59.7 µmol L$^{-1}$ in the Johor River and from 0.3–15.2 µmol L$^{-1}$ in the Johor Strait (Fig. 6a). There was no clear seasonal variation in the relationship of [$NO_3^-$] to salinity, although some pre- and post-monsoon samples in the Johor River estuary had lower [$NO_3^-$] than the monsoon samples.

In contrast, $NO_2^-$ concentration showed a clear enrichment at salinities between 10–27 in both the Johor River and Johor Strait, reaching concentrations of 5–10 µmol L$^{-1}$ (Fig. 6b). At all stations with salinities > 31, [$NO_2^-$] was < 0.53 µmol L$^{-1}$, reaching as low as 0.014 µmol L$^{-1}$ but always remaining detectable. The highest $NO_2^-$ concentration was found in the pre- and post-monsoon samples in the Johor River, suggesting that there might be a seasonal pattern (Fig. 6b). Moreover, there was a significant correlation be-

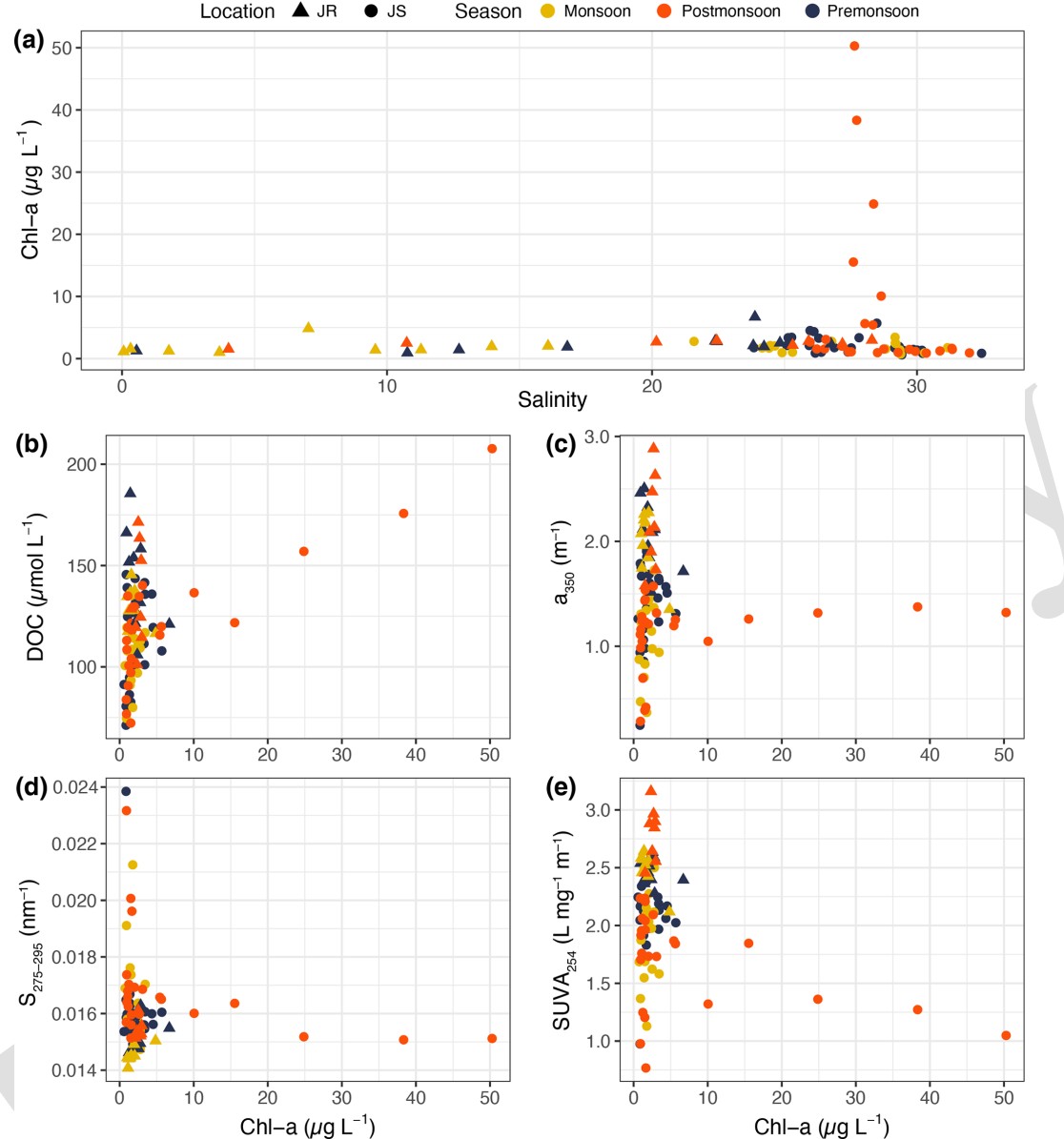

**Figure 4. (a)** Distribution of chlorophyll-*a* concentration against salinity. Scatter plots against chlorophyll-*a* concentration of **(b)** DOC concentration, **(c)** CDOM $a_{350}$, **(d)** the CDOM spectral slope $S_{275-295}$, **(e)** and SUVA$_{254}$. Triangles represent data from the Johor River, circles represent data from the Johor Strait, and symbol colours indicate the sampling season.

tween [NO$_2^-$] and water temperature, although the correlation was fairly weak ($\rho = 0.278$, $n = 94$, $p = 0.008$; Fig. 6c).

The concentration of NH$_4^+$ generally decreased with salinity in the Johor River ($\rho = -0.83$, $p < 0.001$), with values ranging from undetectable to 39 µmol L$^{-1}$ (Fig. 6d). In the Johor Strait, [NH$_4^+$] was low at salinities $\geq 30$ (ranging from undetected to 2.8 µmol L$^{-1}$) but very variable at salinities between 20–30 (ranging from undetected to 51.5 µmol L$^{-1}$; Fig. 6d), with an overall negative correlation between [NH$_4^+$] and salinity ($\rho = -0.56$, $p < 0.001$). NH$_4^+$ concentrations did not show a seasonal pattern.

PO$_4^{3-}$ concentrations were very low in the Johor River (always $< 0.4$ µmol L$^{-1}$ and in numerous cases $< 0.05$ µmol L$^{-1}$) and even showed a weak positive correlation with salinity ($\rho = 0.47$, $p = 0.01$). In the Johor Strait, [PO$_4^{3-}$] showed a similar pattern to that of [NH$_4^+$], with low concentrations ($< 0.34$ µmol L$^{-1}$) at salinities $\geq 30$ and variable concentrations ($< 0.05$ to 2.7 µmol L$^{-1}$) at salinities between 20–30 (Fig. 6e). There was an overall negative correlation between [PO$_4^{3-}$] and salinity ($\rho = -0.48$, $p < 0.001$). Unlike the strong [NO$_3^-$]–salinity relationship (Fig. 6a), [PO$_4^{3-}$] did not show an overall significant rela-

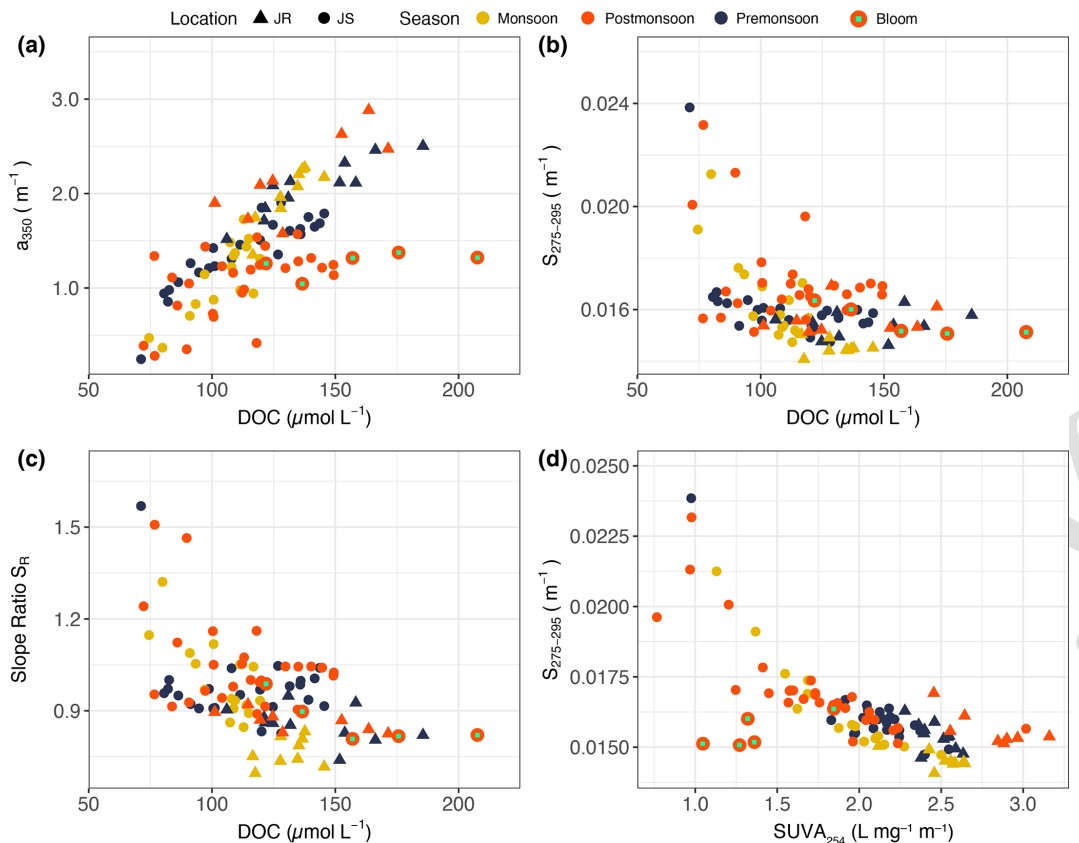

**Figure 5.** Scatter plots showing correlations between **(a)** CDOM $a_{350}$ and DOC concentration, **(b)** the CDOM spectral slope $S_{275-295}$ and DOC concentration, **(c)** the CDOM spectral-slope ratio and DOC concentration, and **(d)** the CDOM spectral slope $S_{275-295}$ and SUVA$_{254}$. Triangles represent data from the Johor River, circles represent data from the Johor Strait, and symbol colours indicate the sampling season. Data from the five bloom stations are indicated by a larger symbol size and a light-green rectangle in the centre of the symbol.

tionship with salinity across the Johor River and Johor Strait. $PO_4^{3-}$ concentration did not show any seasonal variation.

In the Johor River, there was, consequently, no clear DIN–$PO_4^{3-}$ relationship – in fact, a weak negative correlation was seen ($\rho = -0.43$, $p = 0.02$) – and the DIN : $PO_4^{3-}$ ratio was almost always greater than the Redfield ratio of 16 : 1. In contrast, the Johor Strait data did show a linear relationship between DIN concentration and $[PO_4^{3-}]$ ($y = 17.3x + 3.4$, $r^2 = 0.72$, $p < 0.001$), with most data either close to or slightly exceeding the 16 : 1 ratio and a regression slope of 17.3 (Fig. 6f). This suggests an environment in the Johor Strait that is slightly enriched in N compared to P. The relationship was mainly driven by the relationship between $[NH_4^+]$ and $[PO_4^{3-}]$, which followed the Redfield ratio of 16 : 1 fairly closely ($y = 13.3x - 1.5$, $r^2 = 0.73$, $p < 0.001$).

The Si(OH)$_4$ concentration was overall negatively correlated with salinity, reaching values $< 1\,\mu\text{mol}\,\text{L}^{-1}$ at high salinities ($\rho = -0.69$, $p < 0.001$; Fig. 6g). Although the lowest-salinity samples in the Johor River mostly had low $[\text{Si(OH)}_4]$ of $< 10$–$15\,\mu\text{mol}\,\text{L}^{-1}$ (except for one high measurement of 66.5 $\mu\text{mol}\,\text{L}^{-1}$), this is most likely an artefact

caused by silicon polymerisation in low-salinity samples, as discussed in Sect. 4.2. The DIN : Si ratios in the Johor River were mostly between 0.7–3.1, with the lowest-salinity stations reaching 7.1–10, while the salinity in the Johor Strait samples ranged mostly between 0.2–6.0. The highest DIN : Si ratios (12–347) were seen at stations 6–8 in the Johor Strait in February 2018, which were also the stations with highest chl-$a$ concentration ($> 25\,\mu\text{mol}\,\text{L}^{-1}$). Overall, most stations exhibited an excess of DIN relative to the canonical Redfield DIN : Si ratio of 1 : 1. Within the Johor Strait, stations 6–8 typically had higher concentrations of $NO_2^-$ (1.5–4.8-fold higher), $NH_4^+$ (3.9–148-fold higher), and $PO_4^{3-}$ (1.2–12-fold higher) than the other stations.

## 4  Discussion

### 4.1  Sources and mixing pattern of dissolved organic matter

DOC concentrations observed in the Johor River (115–150 $\mu\text{mol}\,\text{L}^{-1}$) are at the lower end of values reported from

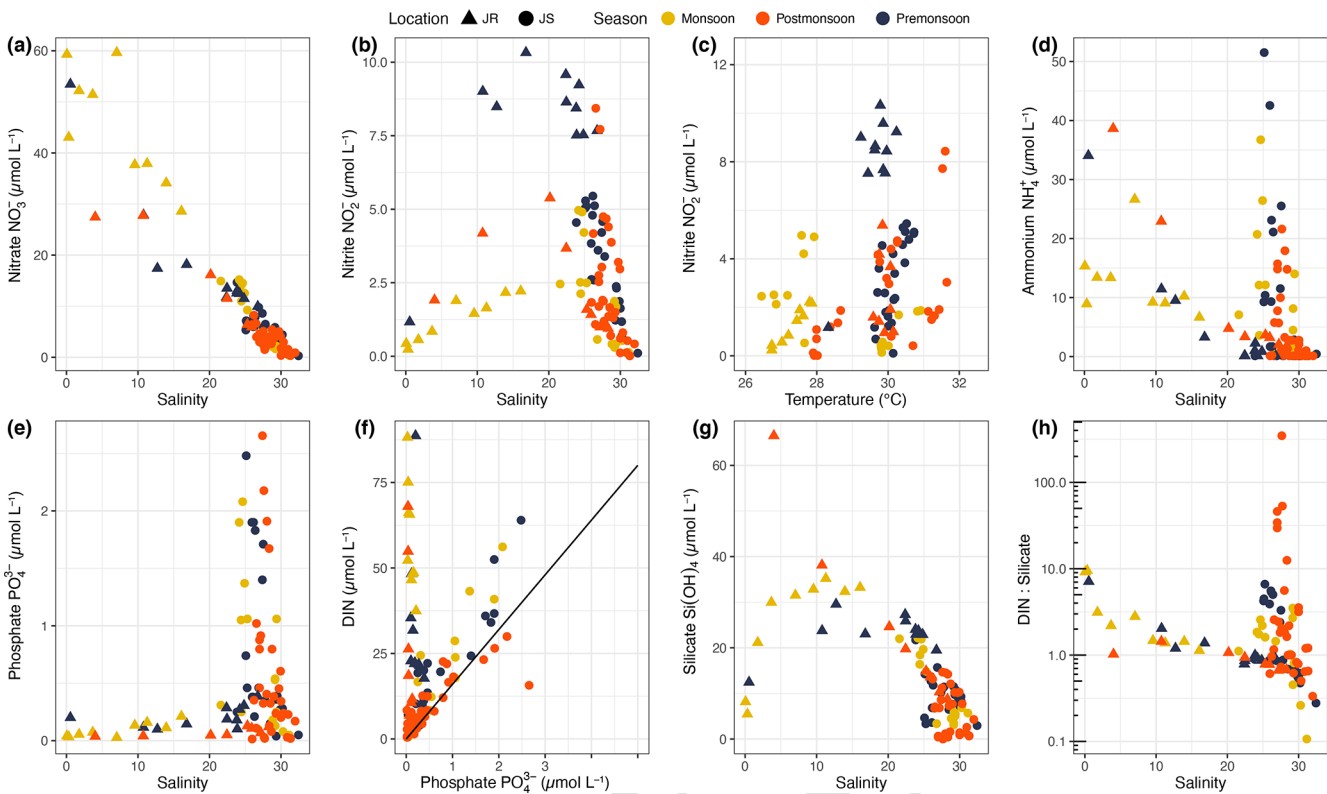

**Figure 6.** Concentrations of **(a)** nitrate and **(b)** nitrite against salinity. **(c)** Nitrite concentration showed a significant correlation with temperature (Spearman's rank correlation). Concentrations of **(d)** ammonium and **(e)** phosphate against salinity. **(f)** Dissolved inorganic nitrogen was generally correlated with phosphate concentration in the Johor Strait but not in the Johor River, where phosphate concentrations were very low. The solid line indicates the 16 : 1 Redfield ratio. **(g)** Concentration of silicate against salinity. The very low silicate concentration for low-salinity Johor River samples is likely an artefact of sample freezing; see discussion in Sect. 4.2. **(h)** Ratio of dissolved inorganic nitrogen to silicate against salinity. In all panels, triangles represent data from the Johor River, circles represent data from the Johor Strait, and symbol colours indicate the sampling season.

river systems in Malaysia and Indonesia (Huang et al., 2017), where rivers that drain tropical peatlands can carry 1000–5000 $\mu$mol L$^{-1}$ (Alkhatib et al., 2007; Wit et al., 2015; Martin et al., 2018). The Johor River values are closer to the values reported upstream of peatlands in the Rajang River ($\sim 120\,\mu$mol L$^{-1}$; Martin et al., 2018) and are consistent with a previously reported wet-season value from the Johor River of 214 $\mu$mol L$^{-1}$ (Huang et al., 2017). Our comparatively low values of CDOM absorption, with $a_{350} < 2.5$ m$^{-1}$ at the lowest salinities in the Johor River, also contrast the much higher values found in the extremely CDOM-rich blackwater rivers found in peatland-draining catchments in Southeast Asia (Martin et al., 2018; Siegel et al., 2019); Africa, e.g. the Congo River and its tributaries (Spencer et al., 2009; Drake et al., 2023); and South America, e.g. the Orinoco River (Battin, 1998). Neither DOC nor $a_{350}$ followed a conservative mixing pattern, with significant DOM addition occurring both within the Johor River estuary and in the Johor Strait.

The low values of $S_{275-295}$ (mostly below 0.016 nm$^{-1}$; Fig. 3d) and $S_R$ (all below 1.0; Fig. 3f) and the elevated SUVA$_{254}$ values (all higher than 2 L mg$^{-1}$ m$^{-1}$; Fig. 3c)

are consistent with a predominantly terrestrial DOM source within the Johor River estuary. These CDOM parameters are used widely to trace terrigenous DOM in estuaries and river-influenced coastal seas (Fichot and Benner, 2011; Lu et al., 2016; Clark and Mannino, 2021). SUVA$_{254}$ is proportional to the DOM aromaticity (Weishaar et al., 2003), and terrigenous DOM typically has high SUVA$_{254}$ with values above 2 L mg$^{-1}$ m$^{-1}$ (Massicotte et al., 2017). SUVA$_{254}$ was also found to be relatively robust at distinguishing terrigenous from aquatic DOM in decomposition experiments (Lee et al., 2018). $S_{275-295}$ is inversely proportional to the apparent molecular weight of DOM (Helms et al., 2008). $S_R$ also correlates with the apparent molecular weight of DOM and has been described as a suitable proxy for distinguishing terrestrial and marine CDOM, with marine-like CDOM having $S_R > 1$ (Helms et al., 2008, 2014). However, Han et al. (2021) found that in a temperate estuary system in Korea, $S_R$ was actually elevated ($\sim 1.3$) in samples richer in terrigenous DOM, which they attributed to prior decomposition. While this indicates that caution is needed when interpreting optical properties of DOM as source indicators, we

note that the relatively low concentrations of chl-*a* in the Johor River (Fig. 4) suggest that autochthonous production is probably not a major source of DOM. Moreover, the high turbidity in the Johor River (total suspended matter concentrations at our stations were typically 10–30 mg L$^{-1}$ (Liang et al., 2020)) makes it unlikely that submerged vegetation or macroalgae contribute substantially to the DOM pool. However, fringing mangroves are found along much of the Johor River estuary, and mangroves are known to provide large inputs of terrigenous DOM (Jennerjahn and Ittekkot, 2002; Dittmar et al., 2006). An input of terrigenous DOM from these fringing mangroves would help to explain the observed non-conservative mixing pattern in the DOC concentration and CDOM properties (Fig. 3).

The values of the CDOM properties were more variable in the Johor Strait than in the Johor River. The low values of $S_{275-295}$ and $S_R$ in most of these samples (below 0.018 nm$^{-1}$ and 1.1, respectively) are typically associated with terrigenous DOM in river-influenced coastal margins (Fichot and Benner, 2011; Carr et al., 2019; Lønborg et al., 2021a; Zhou et al., 2021). The low surface salinities in the Johor Strait found in the present study and reported previously (Gin et al., 2000; Kok and Leong, 2019; Mohd-Din et al., 2020) suggest that terrigenous DOM input from run-off probably does contribute to the DOM pool in the strait, while the fringing mangroves along the strait will also supply terrigenous DOM. However, the SUVA$_{254}$ values were mostly lower in the Johor Strait than in the Johor River estuary at a given salinity, which points to an increasing contribution from autochthonous DOM. Given the eutrophic status of the Johor Strait (Gin et al., 2000; Chénard et al., 2019; Kok and Leong, 2019), autochthonous DOM production is likely substantial. This is not only evident from the high DOC concentrations at the five bloom stations (Fig. 3a and b) but also demonstrated by the bottom-water hypoxia and sedimentary anoxia found in both the West and East Johor straits (Kok and Leong, 2019; Chai et al., 2021). However, the phytoplankton bloom did not appear to contribute much CDOM, given that $a_{350}$ showed little change with chl-*a* concentration (Fig. 4c). This would explain the low SUVA$_{254}$ compared to the bloom stations' $S_{275-295}$ (Fig. 5d), because production of mostly non-coloured DOM by the bloom would lower SUVA$_{254}$ without influencing $S_{275-295}$. Our data therefore suggest that direct production by phytoplankton is probably not a major source of CDOM in the Johor Strait. This leaves production by heterotrophic microbes as a more likely pathway of generating autochthonous CDOM, given that the hypoxia in the inner part of the Johor Strait (Mohd-Din et al., 2020; Chai et al., 2021) clearly indicates that heterotrophic reprocessing of organic matter is substantial. CDOM produced by microbial reprocessing of DOM can have absorbance and fluorescence properties that resemble terrigenous CDOM (Hansen et al., 2016; Osburn et al., 2019).

## 4.2 Nutrient sources and implications for phytoplankton dynamics

Although the $NO_3^-$ concentration showed mixing behaviour that is close to linear across the Johor River and Johor Strait, the pattern of the other nutrients clearly shows that different patterns of nutrient cycling operate in the two locations. Our data indicate that the Johor River is notably enriched in DIN relative to $PO_4^{3-}$. While wastewater treatment plants in the Johor River catchment may be significant point sources of $NH_4^+$ (Pak et al., 2021), $NO_3^-$ probably originates from soil nitrogen, as similarly observed in the Rajang River system in Borneo (Jiang et al., 2019), and possibly from fertiliser use in the largely agricultural catchment. The low $PO_4^{3-}$ concentrations would be consistent with soil nutrient sources, given the predominance of highly weathered Acrisol soils in the catchment (Pak et al., 2021) that are likely to be poor in phosphorus. The concentrations of $NO_3^-$ and $NH_4^+$ are broadly in line with values reported from other river systems in tropical and subtropical Asia with varying degrees of anthropogenic impacts, which can range from tens to hundreds of µmol L$^{-1}$ (Jennerjahn et al., 2004; Cai et al., 2015; Kuo et al., 2017; Suratman et al., 2018; Jiang et al., 2019). The clear increase in $NO_2^-$ concentration at salinities between 10–25 (Fig. 6b) indicates active nitrogen recycling within the Johor River estuary and in the Johor Strait, likely resulting from nitrification of the $NH_4^+$ pool. In a subtropical North American estuary, Schaefer and Hollibaugh (2017) reported that $NO_2^-$ oxidation rates slowed relative to $NH_4^+$ oxidation rates at temperatures of 20–30 °C, leading to a similar accumulation of $NO_2^-$ as in our data. Whether this temperature-dependent mechanism, which uncouples the two steps of nitrification, also applies to permanently warm tropical systems is unknown. Our data do show a significant, albeit weak, correlation between $NO_2^-$ concentration and temperature, which suggests that the temperature sensitivity of biogeochemical rates in tropical estuaries would be an important topic for future research.

The accumulation of $NH_4^+$ and $PO_4^{3-}$ in the Johor Strait, following proportions consistent with the Redfield ratio (Fig. 6d, e, and f), suggests that substantial internal recycling of nutrients takes place in the strait; the importance of recycling is also evident from the accumulation of $NO_2^-$ at lower salinities (Fig. 6b). Elevated concentrations of $NH_4^+$, $PO_4^{3-}$, and $NO_2^-$ were especially noticeable at stations 6–8, closer to the inner Johor Strait. Given that the $NH_4^+ : PO_4^{3-}$ ratios were generally fairly close to 16 : 1 (Fig. 6f), recycling via aerobic respiration is likely important. However, Chai et al. (2021) showed that sedimentary anammox, denitrification, and dissimilatory nitrate reduction to ammonia (DNRA) occur throughout the East and West Johor straits at rates ranging from < 0.5 to 11 µmol kg$^{-1}$ h$^{-1}$ and that these sediments contain a large fraction of iron-bound phosphorus. Although the rate of denitrification + anammox usually ex-

ceeded that of DNRA, the rate of DNRA always exceeded that of anammox and was between 50 % and > 100 % of the denitrification rate at all but one of the stations. Chai et al. (2021) therefore concluded that there is net sedimentary N loss; however, they noted that DNRA also recycles an appreciable fraction of $NO_3^-$ to $NH_4^+$ and that the release of iron-bound P from the sediments might also be important. Our data are consistent with N and P recycling helping to maintain a eutrophic state within the Johor Strait. Further research on pelagic and sedimentary nutrient-recycling rates in this system is therefore warranted.

Tropical rivers typically carry high $Si(OH)_4$ concentrations, averaging close to 200 $\mu mol\,L^{-1}$ in Asia (Jennerjahn et al., 2006). Our low-salinity samples mostly returned values below 30 $\mu mol\,L^{-1}$, yielding a distinct unimodal relationship with salinity. Although it is possible for diatom growth to deplete $Si(OH)_4$ in the freshwater reaches of an estuary (and for dissolution of biogenic silica and of aluminosilicate minerals to be enhanced within the saline reaches of an estuary (Eyre and Balls, 1999; Roubeix et al., 2008a, b)), it is more likely that our result was an analytical artefact caused by silicon polymerisation in frozen samples, which affects low-salinity samples and samples with high $Si(OH)_4$ concentrations more strongly (MacDonald and McLaughlin, 1982). This is expected to be less of a problem in the higher-salinity Johor Strait samples, where $Si(OH)_4$ concentrations were also lower, which showed a much more consistent relationship with salinity (Fig. 6g). Our Johor Strait data indicate a high DIN : Si ratio, typically > 1, which is consistent with previous data from the Johor Strait (Chénard et al., 2019; Kok and Leong, 2019). This contrasts with the consistently low DIN : Si ratios, averaging around 0.3, measured in the Singapore Strait and also using frozen samples (Martin et al., 2022). While the Johor Strait experiences diatom blooms (Mohd-Din et al., 2020; Chai et al., 2021), the excess of N and P relative to Si may favour the growth of non-silicifying phytoplankton, including harmful dinoflagellate blooms that have been observed in the Johor Strait (Kok and Leong, 2019; Chai et al., 2021). Moreover, the fact that DIN : $PO_4^{3-}$ ratios were generally consistent with or above the 16 : 1 Redfield ratio suggests that phosphorus may play a role in limiting phytoplankton production in the Johor Strait. This is consistent with data from Kok and Leong (2019) collected in the East Johor Strait between 2015–2017, which indicated DIN : $PO_4^{3-}$ ratios fairly close to 16 : 1, and more recent measurements taken by Wijaya et al. (2023) in 2020, which revealed DIN : $PO_4^{3-}$ ratios mostly above 16 : 1.

We do not know the taxonomic composition of the phytoplankton blooms encountered in February 2018 (stations 6–8) and June 2018 (station 7). However, on most sampling dates, chl-*a* concentrations were not particularly elevated, despite high concentrations of nutrients (DIN: ;10–50 $\mu mol\,L^{-1}$, $PO_4^{3-}$: 0.5–2.5 $\mu mol\,L^{-1}$). This indicates that factors other than nutrient availability, most likely water column stability, light penetration, and ecological interactions

(such as grazing or viral lysis), are also important controls over bloom formation in the Johor Strait (Chen et al., 2009; Davidson et al., 2014). The fact that one bloom was observed in February 2018, when rainfall was low and the salinity at the bloom stations was relatively high (27.6–28.7 compared to an overall range of 24.2–29.4 at these three stations across the sampling period), also suggests that direct freshwater run-off was probably not in itself a key trigger for the bloom. Our data are consistent with a recent analysis of microbial-community variation over 2 months in late 2020, which concluded that phytoplankton biomass in the East Johor Strait is likely under a significant degree of top-down control (Wijaya et al., 2023).

## 4.3   Seasonal biogeochemical variation

Although rainfall and therefore river discharge showed clear seasonal variation, this did not result in strong seasonal variation in the biogeochemical parameters we measured, which were consistent with previous studies in the Johor Strait (Chénard et al., 2019; Mohd-Din et al., 2020). This is despite the fact that the DOM pool in the Johor River is likely largely of terrigenous origin (based on optical properties), and terrestrial DOC concentrations are typically expected to increase with river discharge (Raymond and Saiers, 2010; Kurek et al., 2022; Drake et al., 2023). The somewhat lower DOC concentrations in the Johor River estuary during the northeast monsoon might indicate a dilution effect due to increased rainfall, as observed in peatlands (Clark et al., 2007; Rixen et al., 2016). This was not observed for $NO_3^-$, possibly indicating that $NO_3^-$ originates from shallower horizons of the soil profile than DOC. However, our sampling in the Johor River was limited to three dates and did not capture periods of peak river discharge (Fig. 2b); thus, the true seasonality of riverine concentrations may have been missed. The only parameter for which some seasonality was apparent in our data was $NO_2^-$, which we hypothesise might be linked to temperature, as discussed above in Sect. 4.2. The Johor River estuary and Johor Strait are thus different from estuaries located in seasonal wet–dry tropical climates, where large seasonal variation in precipitation causes much more pronounced seasonal variation in biogeochemistry (Eyre and Balls, 1999; Pratihary et al., 2009; Burford et al., 2012). Our data also suggest that dissolved nutrient concentrations in the eutrophic Johor Strait are not primarily controlled by seasonal precipitation and run-off patterns and that phytoplankton blooms in the strait are not purely controlled by nutrient availability.

While the seasonal variation in rainfall reflected the long-term average of seasonality fairly well, we acknowledge that our study only encompassed 1 year, and thus the interannual variation is at present unknown. The hydroclimate across Southeast Asia is influenced by both the El Niño–Southern Oscillation (ENSO) and the Indian Ocean Dipole (IOD) (Xiao et al., 2022), although a previous analysis of river discharge data concluded that ENSO alone did not sig-

nificantly influence the variability in Johor River discharge (Xu et al., 2004). We therefore expect that the main conclusions reached in the present study are probably generalisable across years, but more research is needed to resolve longer-scale temporal variability. Any additional anthropogenic impacts and land-use changes are also likely to further alter biogeochemical cycling in this system, and our data thus provide a valuable baseline against which future data can be compared.

The limited seasonal variability observed in the present study contrasts with the Singapore Strait directly to the south, where the monsoonal reversal of the prevailing ocean currents delivers a seasonal input of terrigenous DOM and nutrients during the southwest monsoon (Zhou et al., 2021; Martin et al., 2022). Our data further show that the Johor River carries much lower concentrations of DOC but much higher concentrations of $NO_3^-$ compared to the inferred concentrations of the river input that affects the Singapore Strait – around $900\,\mu\mathrm{mol\,L^{-1}}$ for DOC (Zhou et al., 2021; Chen et al., 2023) and around $24\,\mu\mathrm{mol\,L^{-1}}$ for $NO_3^-$ (Martin et al., 2022). This further supports the conclusion that input from the Johor River has little impact on the biogeochemistry of the Singapore Strait (Zhou et al., 2021; Martin et al., 2022), as also shown for suspended-sediment concentrations by van Maren et al. (2014). Based on the seasonality and inferred riverine endmember concentrations, the terrigenous input to the Singapore Strait is instead derived from peatland-draining rivers (Zhou et al., 2021; Martin et al., 2022).

## 5  Conclusion

The Johor River and Johor Strait are clearly biogeochemically distinct and not simply part of the same estuarine mixing continuum. The Johor River appears to carry mostly terrigenous DOM, albeit with significant non-conservative additions within the estuary. In the eutrophic Johor Strait, phytoplankton blooms produce autochthonous DOM. Although there were lower $SUVA_{254}$ values in the Johor Strait than in the Johor River, the parameters of the CDOM spectral slope in the Johor Strait were consistent with typical terrigenous values, demonstrating that optical properties of CDOM may be ambiguous source indicators in eutrophic waters, where heterotrophic microbes likely produce CDOM as well. Our data further reveal possible evidence for temperature-dependent $NO_2^-$ accumulation in estuarine waters despite the limited range in seasonal temperature. The large contribution of $NH_4^+$ to the DIN pool, with $DIN:PO_4^{3-}$ ratios generally at or above $16:1$, indicates that internal nutrient recycling is likely important in the Johor Strait and that phosphorus may at times be the limiting nutrient in this system.

*Data availability.* All raw data and analysis codes are available via the Nanyang Technological University (NTU) data repository at https://doi.org/10.21979/N9/XJWPHI (Martin, 2023).

*Author contributions.* Conceptualisation: PM, AYLC, KVA, and ELY. Funding acquisition: PM. Investigation: AYLC, PM, KVA, YZ, and RSN. Formal analysis: AYLC and PM. Supervision: PM. Writing (original draft): AYLC. Writing (review and editing): AYLC, PM, KVA, ELY, YZ, and RSN.

*Competing interests.* The contact author has declared that none of the authors has any competing interests.

ther geographical representation in this paper. While Copernicus Publications makes every effort to include appropriate place names, the final responsibility lies with the authors.

*Acknowledgements.* We are grateful to Chen Shuang and Ashleen Tan Su Ying for their help with sample analysis and fieldwork and to Anne Leong and Tan Keng Meng for logistical support during fieldwork. We thank the two anonymous reviewers for their constructive feedback, which improved the original paper.

*Financial support.* This work was funded through the Singapore Ministry of Education Academic Research Fund Tier 2 (grant no. MOE-MOET2EP10121-0007) and through a Nanyang Technological University start-up grant.

*Review statement.* This paper was edited by Yuan Shen and reviewed by two anonymous referees.

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

**Remarks from the language copy-editor**

CE1    Please note the slight changes made to the third and fourth affiliations, as these might not appear in the track-changes file.

CE2    Please note the similar change that has been made to the title of the paper.

**Remarks from the typesetter**

TS1    Please provide a short explanation regarding this correction that can be forwarded by us to the editor. Changes in values require editor approval. Thank you very much in advance for your help.