# Peer review of "Distribution of nutrients and dissolved organic matter in a eutrophic equatorial estuary, the Johor River and East Johor Strait"

_EGUsphere, 2023_

## Author Response (AR1)

**Response to Reviewer 1**

**General comments**

Cheong et al. present monthly sampling results for nutrients, dissolved organic carbon, and chromophoric DOM in a tropical estuary over the course of one year. This study advances our understanding of biogeochemical cycling in tropical, eutrophic estuaries receiving inputs from non-peatland draining rivers, which are understudied systems.

The approach is generally sound and the questions are relevant to the biogeochemical community. The conclusions are meaningful, overall supported by results and integrating related work in the estuary.

The writing lacks specificity at times and statements could be better supported by statistical analyses or with more precise language.

> We thank the reviewer again for their appreciative feedback. The we have improved the relevant passages by making the language more specific and adding statistical analysis where needed, as outlined below. We have also carefully re-read the manuscript to edit the language for specificity, conciseness, and consistency.

**Specific comments**

It is unclear why the November 2017 sampling was excluded from the monsoon period (Line 108), when rainfall and streamflow data (Fig. 2 and section 3.1) show that this month is part of the wettest part of the year? You also state that the early northeast monsoon is from November to early January on Line 103. This is the only questionable aspect of the approach, but it is a substantial concern that may require a thorough re-analysis of the dataset after re-classification of the Nov. 2017 data points.

> We had not explained this very clearly: the NE monsoon in fact typically begins in mid-November, and our first sampling trip in the Johor River was on 3$^{rd}$ and 4$^{th}$ November 2017, which was before the river discharge started to increase with the NE monsoon rain. We have clarified this point now in Sections 2.1 and 2.2, and have given the exact sampling dates for the Johor River sampling. In Figure 2 we have also indicated the sampling dates on the panel with river discharge, so it can be clearly seen why we designate the 3$^{rd}$–4$^{th}$ November sampling as "pre-monsoon". Given that the river discharge had not yet started to increase, we believe that our original designation was appropriate and we have therefore not changed the analysis.

The correlation between $NO_2^-$ and water temperature (Lines 263-264) may be significant but it is weak (rho = 0.278). This result statement needs to be more moderate and needs to acknowledge that the correlation is weak. This note is particularly important because you refer to the significant but weak correlation in the Discussion (Line 363). Please moderate this statement as well.

> We agree that this correlation is weak and therefore should not be over-interpreted. We have now acknowledged in both the Results and Discussion that the correlation is weak. We're not trying to draw strong conclusions from this correlation anyway, and are simply highlighting this to make the case that the temperature-sensitivities of rate processes would merit further investigation.

There are several instances where statements need to be better supported with statistical tests or with the use of more quantitative terms. I highlighted the more specific instances in the next section (Technical corrections), but the most pressing needs for quantitative support are in Section 3.5. Please quantify the linear relationships mentioned; nitrate-salinity, lack of phosphate-salinity relationship, DIN-phosphate in the Johor Strait. Similarly, Lines 230-234, when describing linear relationships, could you give the $R^2$ and p values of the linear regression to support the statements that there is a linear relationship? It is evident for DOC but it is less clear that there is a linear relationship for SUVA and Chl-a. Back this up a little bit, at a minimum with the characteristics of the linear regression. Line 297: is there a way to support this statement quantitatively/with statistics?

> We have now calculated these relationships and correlations and provided the statistical information wherever relevant in the text. Specifically, we have examined the correlations between DOC and chl-a in more detail and provided the statistical information in Section 3.3. We have also examined the relationships in the nutrient concentrations in more detail and provided the statistical information in Section 3.5. For the last sentence of the results (prev. Line 297), we have now calculated the ratio of the average nutrient concentrations at Stations 6–8 to the average nutrient concentrations at the other five stations, and refer to this result in that sentence.

Writing lacks specificity at times. For example, Lines 427-428; modulate this statement to indicate that this is a conclusion based on this study, not a general truth. A common issue is the need to specify that you are referring to concentrations when you present nutrient results. Below are a few instances, others are detailed in the next section:

- Line 256: specify "concentrations" after nitrate, or use concentration brackets. Similar comments throughout the entire section 3.5

- Line 310: "low" and "elevated" compared to what? Whenever using high or low, a reference or comparison point needs to be provided.

Please pay better attention to units. There are many instances where units are missing or incorrect. All issues are listed in the next section, but together they add up to enough concern as they minimize the impacts of your results.

We have edited the text so that it is always specified that we are talking about concentrations wherever relevant. We have also checked for units, although in the case of salinity we continue to report unitless values (please see our response to the comment on salinity units below). We have also tried to avoid using comparative terms ("high", "low") without specifying what this is relative to.

The abstract should mention or better highlight two key conclusions from section 5 (Conclusion): "The Johor River and Johor Strait are clearly biogeochemically distinct and not simply part of the same estuarine mixing continuum" and "CDOM optical properties may be ambiguous source indicators in eutrophic waters where heterotrophic microbes are likely producing CDOM as well."

We have added these conclusions to the abstract.

The description of the study area (2.1), Figure 1, and section 3.1 assume that the reader is familiar with the area. While I enjoyed learning more about the local geography and toponomy, not every reader will. Please ensure that the information provided is easy to grasp by addressing the following issues:

- It is essential to show Singapore on this map since it is mentioned in Methods and Discussion. Similarly, the East Johor Strait, the Singapore Strait, and Pulau Ubin need to be labeled.
- 1: please show the land use for Singapore, if possible, as the East Johor Strait receives water inputs from this land too.
- Lines 93-94: please label the East and West portions of the Johor Strait in Fig. 1, we can't follow this sentence without a labeled map.
- Line 184: how is the reader supposed to know where Kota Tinggi is located without a reference to the map and without a mention of the town on the map? Modify the map or change text to help the reader follow.

We agree that the map was not sufficiently clear, as also highlighted by Reviewer 2. We have added the land-use for Singapore and have added all necessary place names and labels that are referred to in the text.

Lines 129-132: please provide a detection limit for the DOC analysis to parallel the nutrient analyses paragraphs.

> Detection limits are never really an issue for estuarine and marine DOC analysis because the instrumental detection limit is considerably lower than even the lowest DOC concentrations found in the deep ocean. We use a lowest calibration standard of about 20 µmol/l DOC, and the LOD is lower than this (probably somewhere in the region of 10–15 µmol/l), and thus much below the lowest measured concentration in this dataset of 71 µmol/l. We have added this information in Section 2.3.1.

Line 246: the five bloom stations are not obvious in Fig. 5d, please either circle them on the figure or remind us where to look (low $S_{275-295}$, low SUVA). These stations are referred to in the Discussion, so it is an important note that all readers should easily grasp.

> We have now highlighted the five bloom stations in all four panels of Fig 5. Because they don't all cluster completely separately from the other stations in all panels, we have decided to highlight them by increasing the symbol size and placing a small centre rectangle in light green colour in the centre of the plot symbol. This has also been added to the figure legend.

In the discussion of seasonal variations in biogeochemistry, please acknowledge that this study was limited to one year, thus potentially missed inter-annual variability in precipitation and streamflow. The long-term mean monthly precipitation is helpful to justify extrapolating the study findings to other years, but year 2017-2018 nonetheless had its unique precipitation and discharge patterns that may not match mean trends. The inter-annual hydrologic variability of the system should be acknowledged, at a minimum, or discussed in more detail, ideally.

> We agree that it is important to acknowledge the limitations of only having sampled in one year. We have added some discussion as a new second paragraph of Section 4.3 to point out that interannual variation is currently unknown. The obvious candidates for climatically-driven variability would be ENSO and IOD, as we now point out there, although their influence may be weaker than in other parts of Southeast Asia. Since the relationships between precipitation, run-off/discharge, and inputs of nutrients and DOM are not well known in this region, we don't wish to speculate too much about the possible directions of future changes. We also point out that any further anthropogenic changes would be likely to impact this system, so our data are also valuable as comparison for any future research in this location.

I was going to suggest citing Spencer et al. (2009), another biogeochemical assessment of a tropical river (the Congo River), but I noticed it was listed in the References. However, it does not appear in the manuscript. Two comments:

- Please do cite this study and compare its findings to your findings.
- Double-check that the list of cited references to make sure all cited works are listed and vice versa.

We have now properly included the reference to Spencer et al. 2009 in Section 4.1. We have also added a few other relevant citations and we have double-checked that all cited references and the reference list match properly.

**Technical corrections**

Line 23: specify salinity units.

As mentioned in our previous response, salinity data based on the EOS-80 equation of state are unitless conductivity ratios that are expressed on the practical salinity scale, and the use of "psu" as a unit is discouraged (e.g. see Millero 1993 [https://tos.org/oceanography/assets/docs/6-3_letter.pdf], Pawlowicz et al. 2012 [https://core.ac.uk/download/pdf/42147716.pdf], also Pawlowicz 2013 [https://www.nature.com/scitable/knowledge/library/key-physical-variables-in-the-ocean-temperature-102805293/]). We have therefore decided against adding "psu" to the axis labels or using this in the text.

Line 137 already specified that our salinity is reported on the practical salinity scale, and we have added that the salinity is therefore unitless.

Lines 23, 24, 25: Specify "concentrations" after ammonium and phosphate.

Done.

Line 34: remove the repetition of important/importance, consider using value instead of importance.

We have changed "importance" to "value", we agree that this reads much more elegantly.

Line 46: add "the" in front of release.

Done.

Line 47: define (spell out) anammox.

Done.

Line 49: improve clarity by rewording to "differ between tropical and temperate systems".

Done.

Line 55: CDOM is defined as chromophoric DOM in the abstract. Ensure consistency in CDOM definition between the abstract and the main text.

We've changed the definition in the abstract to match the text

Line 68: "behavior" should be plural.

Done.

Fig 1: I suggest swapping the order of maps in panels B and C so the river stations are on top and next to the Johor River on the main map and the Strait stations are at the bottom, closer to their symbols on the main map.

Fig 1: the inset map in panel A should have a better definition; when zoomed in to read it, the letters appear unclear.

Fig 1: it may be helpful to specify what kampung means in parentheses? Add (village) after Kampung in the legend.

These three comments have been taken over for the new version of Figure 1.

Line 95: start a new sentence at fringing mangroves.

Done.

Line 161 and section 3.2: units (psu) need to be shown for salinity values.

As per our response above, we have decided to avoid using psu.

Fig.3c, Lines 212-213, Fig.5d: SUVA units are missing the liter component: l mg$^{-1}$ m$^{-1}$.

Units have been added to the figure.

Line 213: incorrect unit for spectral slope (typo).

The units for the spectral slope are correct here in the text (they were wrong in Figure 3 but have been updated), they indicate the change in CDOM per nano-metre of wavelength, hence units of nm$^{-1}$.

Line 218: remove "values of" to avoid repetition.

Done.

Line 219: remove "especially", it conflicts with "while".

Done.

Line 226 (and general comment for the entire manuscript): add the year after each month to help the reader remember when the study took place.

Done here and throughout the manuscript

Fig.4: units are missing for several parameters.

Units have been added.

Figure 5: why are the rho values shown in parentheses preceded by a c? c(...)?

Apologies, this way a typo and not intended. We have decided to remove the statistical information from the figure panels in Fig. 5 and 6 because we now report a larger number of statistical tests, and in several cases we report separate statistical information for the Johor River and the Johor Strait data. It would be hard to include all of this information on the figure panels, and hence we decided to be consistent and report the statistical information only in the text.

Lines 257, 273: fix the hyphenation (currently shown as a superscript).

This part has been rephrased to avoid a hyphen following a superscript minus sign.

Fig.6d: ammonia is $NH_3$, this is ammonium (apply change to the entire manuscript).

Fig.6f: DIN needs units.

Both changes have been made in Figure 6, and we have corrected to "ammonium" throughout the MS.

Line 271: use plural for PO4 concentrations.

We've switched here to using concentration brackets instead.

Line 293: add "values" after salinity or use salinities (plural).

This paragraph has been slightly rewritten now, and this change is taken care of.

Line 303: I would recommend rewording to "Johor River values".

Done.

Line 310: "low" and "elevated" compared to what? Specify.

Line 313: similarly, what is "high SUVA"?

For both these comments, we have now provided better context.

Line 328: please consider rephrasing this topic sentence to make it more active and concise.

Rephrased to "*The values of the CDOM properties were more variable in the Johor Strait compared to the Johor River*".

Line 329: units are missing for $S_{275-295}$.

Units have been added.

Line 338: we tend to limit the use of "significant" for statistics. I suggest a rephrase.

Agreed, we have rephrased to "*did not appear to contribute much CDOM*".

Line 365: high compared to other systems? Compared to what? It may be more rigorous to remove "high" and simply comment on the accumulation.

We've removed "high".

Line 385: please refer to Fig.6g to help the reader follow.

We've added the reference to Fig. 6g.

Line 396: please remind us of the year to make this study meaningful in a few years and to help with clarity.

We've now added the year wherever we have referred to months.

Line 412: a word is missing after shallower.

Rephrased to "*from shallower horizons of the soil profile*".

Line 426: clean up the citation.

Done.

Line 432: the Johor Strait is not a person, doesn't have eyes, can't see. Please rephrase.
Rephrased to "*In the eutrophic Johor Strait, phytoplankton blooms produce autochthonous DOM*".

**Response to Reviewer 2**

We thank Reviewer 2 for their time and constructive comments. We will address each of these comments as detailed below.

This is a nice case study on DOM and nutrients cycling in a river estuariy and a strait in the tropic. The data obtained from less-well studied tropical region are valuable and will help the scientific community to better understand the global cycling and budget of DOM. The manuscript is well written. The science appears sound and robust. I have only minor suggestions to improve the manuscript.

The data are only presented and interpreted over the salinity gradient. The majority of the data are in the high salinity and only limited data are in the lower salinity area. It will be helpful if the data are presented as contour plots on the map especially for the river data. Or another way is to provide salinity contour plot on the map of the river. Readers may want to see where the 25 per mil salinity boarder is located on the map although it may vary depending on the monsoon. This way, readers can better see where the shift from the constant DOC concentration to decrease with increasing salinity begins geographically.

Because our sampling could not be conducted along an evenly spaced section or grid, we have decided instead to simply visualise the salinity in the Johor River estuary as coloured points for the sampling sites. This gives the reader an idea of the spatial variability in salinity without extrapolating too much from our data across the rest of the estuary. These figures are provided as supplementary information and are referred to in Section 3.2.

Figure 1. It will be good to have goegraphic names such as Pulau Ubin are indicated on the map.

As also requested by Reviewer 1, we have improved Figure 1, and all geographic names are now properly indicated.

Full names of abbreviations such as GPM IMERG should also be provided.

Full name has now been provided.

Figure 2: It will be better to make the x-axes of plots a and b match.

The x-axes are now matched in Figure 2.

---

## Author Response (AR2)

Dear Dr. Shen,

Thank you for further reviewing our revised submission. We are very pleased to hear that the new version was evaluated positively. We have addressed your additional minor comments as follows:

I have reviewed the revisions and have a few more suggestions for you.

Abstract: I recommend adding 1-2 sentences at the end summarizing the conclusions and implications of the major findings, as well as the broader impact of this work.

We have changed the order of the final two sentences of the abstract, and added on the following text:

> *Overall, our results indicate that the Johor River and Johor Strait are clearly not part of the same estuarine mixing continuum, and that nutrient recycling processes must be quantified to understand nutrient dynamics in the Johor Strait. Moreover, our results highlight the need for better DOM source tracing techniques in eutrophic estuaries.*

Our study is ultimately a site-specific investigation, so we don't want to reach too far beyond our data to draw overly broad general conclusions. Hopefully our addition is sufficient. We do believe that our study is a very important starting point for further investigations of this system, which we are planning as part of a new proposal in which we intend to measure nutrient recycling rates and would start to set up a biogeochemical model to work towards climate change projections. The data in the current study are very important to help shape the direction of this research, and we hope that further investigations of this system will help contribute more general knowledge about tropical estuarine biogeochemistry.

Line 61: It would be beneficial to include a few references to support the statement of "Tropical peatlands are the largest source..."

We have added a few relevant references for this sentence.

Lines 65-66: It would be helpful to provide a brief overview of previous studies on the distributions of DOM and nutrients, including their findings and remaining questions. This would strengthen the motivation for this work.

We have added a new paragraph citing some additional relevant literature on urbanised estuaries in Southeast Asia. A detailed review would be beyond the scope of our introduction but the Tanaka et al. (2021) paper (which we cite there) provides this for interested readers. An important point that we now make in this new paragraph is that the biogeochemistry of different estuaries is strongly dependent on site-specific factors, as shown clearly in the Tanaka et al. review, and this provides a clearer motivation for our study: the system we are

investigating here has not been studied in any detail for its biogeochemistry before, so our study provides an important starting point.

Figure 2 appears to be of low resolution, making it difficult to see the axis. Could the authors provide a higher resolution version?

Apologies for that. The issue is to do with the embedding of the figure file in the Word document, which in this particular case leads to a less than optimal resolution. The original figure is a pdf vector graphic that does not loose resolution, and we will submit the original vector file for this and all other figures after acceptance, so the final published manuscript will not have this problem. In the new version, it seems that the embedded file is now also coming out with good resolution.

Looking forward to receiving your revisions.

Best regards,
Yuan Shen

Thanks again to you and the reviewers for your time and support.

Yours sincerely,
Patrick Martin
(on behalf of all co-authors)